# Improving Protein–Ligand Interaction Modeling with cryo-EM Data, Templates, and Deep Learning in 2021 Ligand Model Challenge

**DOI:** 10.3390/biom13010132

**Published:** 2023-01-09

**Authors:** Nabin Giri, Jianlin Cheng

**Affiliations:** Department of Electrical Engineering and Computer Science, University of Missouri, Columbia, MO 65211, USA; ngzvh@missouri.edu

**Keywords:** ligand challenge, cryo-EM, protein–ligand interaction, bioinformatics, machine learning, deep learning

## Abstract

Elucidating protein–ligand interaction is crucial for studying the function of proteins and compounds in an organism and critical for drug discovery and design. The problem of protein–ligand interaction is traditionally tackled by molecular docking and simulation, which is based on physical forces and statistical potentials and cannot effectively leverage cryo-EM data and existing protein structural information in the protein–ligand modeling process. In this work, we developed a deep learning bioinformatics pipeline (DeepProLigand) to predict protein–ligand interactions from cryo-EM density maps of proteins and ligands. DeepProLigand first uses a deep learning method to predict the structure of proteins from cryo-EM maps, which is averaged with a reference (template) structure of the proteins to produce a combined structure to add ligands. The ligands are then identified and added into the structure to generate a protein–ligand complex structure, which is further refined. The method based on the deep learning prediction and template-based modeling was blindly tested in the 2021 EMDataResource Ligand Challenge and was ranked first in fitting ligands to cryo-EM density maps. These results demonstrate that the deep learning bioinformatics approach is a promising direction for modeling protein–ligand interactions on cryo-EM data using prior structural information.

## 1. Introduction

Proteins are a building block of life and carry out many vital biological functions. Whether acting as an enzyme to accelerate the chemical reactions, or as regulatory molecules binding to other molecules to activate their functions, the detailed characterization of proteins and their interaction with their binding partners (e.g., the natural substrates or drugs as ligands) is of great importance. Protein–ligand interactions are necessary requirements for signal transduction, immune responses, and gene regulation in living organisms. The study of protein–ligand interactions is important in understanding the mechanisms of biological regulation and provides a theoretical basis for the design and discovery of new drugs. A fundamental objective of computational structural biology is to understand and model such molecular interactions of living systems in sufficient detail so that the behavior of the system can be predicted or modified as desired. In order to characterize the thermodynamic and kinetic behavior of components and their interactions of living organisms, an image of interacting molecules, such as protein–ligand complexes, at near atomic resolution is required to analyze and understand the physical and geometrical constrains of the molecules.

cryo-EM, an acronym for the cryogenic electron microscopy technique [1], is a revolutionary technology that enables the determination of a 3D structure of macro-molecular complexes at atomic resolution. With the development of various techniques in the cryo-EM realm to generate high resolution maps, as seen in Figure 1, EMDataResource [2] has seen a surge in the deposition of cryo-EM derived protein density maps which elucidate the protein and ligand interactions in the molecules. The EMDataResource 2021 Ligand Model Challenge [3] was hosted to rigorously benchmark the current methods for generating models using cryo-EM density maps to improve the prediction and validation of protein and ligand interactions, and to identify the metrics which are most suitable for comparing the fit of atomic coordinate models into the cryo-EM maps.

One of the most popular approaches to modeling the protein–ligand complexes is molecular docking [4,5,6,7], which uses physics- or statistical potential-based molecular simulations to generate protein–ligand complex models and a scoring function for the estimation of their binding affinities to rank them. With the recent advancement in the field of deep learning, another most prominent approach to modeling protein–ligand complexes is deep learning-based methods. Deep learning-based methods predict protein–ligand binding sites [8,9,10,11] using various neural network architectures such as convolution neural networks (CNN), long short-term memory networks (LSTM), and residual networks (ResNet). These methods primarily use three databases: BioLiP [12], ATPBind [13] and Sc-PDB [14] to train and validate their deep learning models before making binding-site predictions. Similarly, deep learning architectures, such as CNNs, graph neural networks, and attention mechanisms, are used for the prediction of protein–ligand binding affinity [15,16,17,18,19,20,21,22,23,24,25,26,27]. These methods mainly make use of two databases: PDBbind [28] and the CASF databases [29] for binding affinity predictions. More advanced methods such as Equibind [30]—an SE(3)-equivariant geometric deep learning model for direct-shot prediction of receptor binding location and ligand’s bound pose—and DIFFDOCK [31]—a diffusion generative model tailored to the task of molecular docking—have been developed recently. However, even with significant research efforts, despite some success, the protein–ligand interaction prediction problem still remains unsolved because existing methods cannot leverage vast structural data effectively.

**Figure 1 biomolecules-13-00132-f001:**
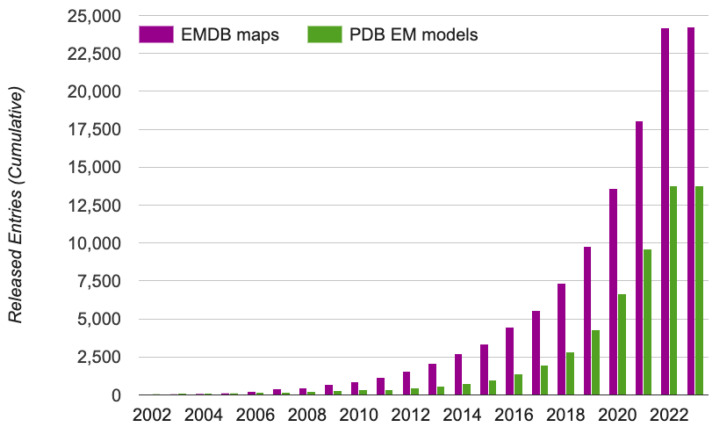
The growth of cryo-EM density maps and cryo-EM-derived protein structures. The statistics were obtained from EMDataResource [32], a unified data resource for 3-Dimension electron microscopy (3DEM) on 20 November 2022.

Inspired by the success of AlphaFold [33], which uses a novel deep learning architecture to predict the protein structures using amino acid sequence data as an input, as well as the various deep learning-based protein–ligand modeling techniques, we adapted the deep learning-based approach for our work. In this work, we combined the deep learning-based protein structure prediction tool DeepTracer with template-based protein–ligand interaction prediction in order to determine the structures of protein–ligand complexes for the 2021 Ligand Model Challenge that was held from 1 February to 1 April 2021. Based on the official results provided by the assessors of the challenge, our method performed best in fitting ligands to cryo-EM maps (measured across all targets), demonstrating the unique value of the novel deep learning bioinformatics approach for modeling protein–ligand interaction.

## 2. Materials and Methods

We attempted to solve the problem of protein–ligand interaction by using a set of bioinformatics methods, incorporating cryo-EM data and known structural information such as reference protein structures. In particular, we leveraged the recent advance of applying deep learning to directly predict the structure of proteins from high-resolution cryo-EM density maps; a succinct review of the methods can be found in Ref. [34]. To predict the bound conformation (3D atomic structure) of a protein–ligand complex, we utilized an existing deep learning-based tool as a key component of our model building pipeline (DeepProLigand). DeepProLigand predicts the 3D coordinates of protein structures using only a cryo-EM density map as an input. This protein structure model is a starting point for the downstream ligand positioning and model refinement tasks. The workflow illustrated in Figure 2 demonstrates our approach to generating the structure of a protein complex by incorporating a fully automatic deep learning-based method as its primary building block. The modeling pipeline of Figure 2 has three key steps described as follows:

### 2.1. Protein Complex Reconstruction from cryo-EM Density Maps and Reference Structures

Using DeepTracer [36], we first predicted the 3D backbone coordinates of the protein complex directly from a cryo-EM density map. DeepTracer uses a 3D U-Net architecture which is modified from the original 2D U-Net [37] architecture developed for biomedical image segmentation. The output from the DeepTracer block is a predicted 3D backbone coordinate structure that has the carbon, carbon alpha, nitrogen and oxygen atoms in the Protein Data Bank Format (PDB), which is standardized by wwPDB [38]. The predicted structure reflects the conformation of the protein in the ligand binding mode. Because the reference structure of the protein (prior structure without cryo-EM information) is also provided by 2021 Ligand Model Challenge organizers, we used the structural alignment to combine them to generate a posterior structure, conceptually similar to combining the prior probability and likelihood to generate a posterior probability in the Bayesian reasoning. Specifically, in order to combine the reference structure and the predicted structure together in terms of geometrical alignment, we utilized the UCSF Chimera’s [39] matchmaker function to superimpose both structures together. Once the structures were superimposed, we saved the superimposed structure relative to reference structure into a new PDB file. The new PDB file then contained the atoms of both reference and predicted structures in the same geometrical space, allowing us to average the coordinates of the corresponding backbone atoms and utilize the reference structure’s residue, and chain labeling for all the shared components between the two structures. The side chains were added on top of the combined backbone structures using the SCWRL4 [40] tool. Finally, a full-atom combined structure, consisting of multiple chains, was produced for the downstream processing. It is worth noting that our approach of generating protein complex structures was different from the traditional approach of fitting a reference structure into a cryo-EM density map.

Algorithm 1 depicts the pseudo code of averaging the backbone atoms’ coordinates of reference and predicted structures. We started by initializing an empty PDB file named “average structure” that followed the guidelines of wwPDB [38]. For each residue of the reference structure, if the distance between the reference structure’s carbon alpha atom and any carbon alpha atom of predicted backbone structure in the 3D geometrical space was less than a threshold value, then all the backbone atoms’ coordinates of the residue in the particular reference structure were averaged with the predicted structure’s corresponding residue and saved in the average structure PDB file; otherwise, the coordinates of the residue in reference structure were simply saved in the average structure PDB file. We refer to the arithmetic mean, the sum of collection of numbers divided by the count of numbers, as “average” throughout the paper. The default distance threshold value we used is 1 Angstrom (Å); however, the threshold value for each chain can be modified as desired.

(1)
$$distance=\sqrt{{\left(x_{r}-x_{p}\right)}^{2}+{\left(y_{r}-y_{p}\right)}^{2}+{\left(z_{r}-z_{p}\right)}^{2}},$$


(2)
threshold=1Angstrom.


Here, let 
$x_{r}$
, 
$y_{r}$
, and 
$z_{r}$
 be the coordinates for each carbon alpha residue of the reference structure and 
$x_{p}$
, 
$y_{p}$
, and 
$z_{p}$
 be the coordinates for each carbon alpha residue of the predicted structure. With this notation, we followed Algorithm 1 and generated new coordinates 
$x_{avg}$
, 
$y_{avg}$
, and 
$x_{avg}$
 which are the average coordinates of both reference and predicted structures. After computing the distance using Equation (Equation 1), we used a threshold value as shown in Equation (Equation 2) for Target 202 and Target 203 of the 2021 EMDataResource Ligand Challenge. For Target 201 of the challenge, since most of the chains were turned into coils, we used a threshold value of 0.3 Angstrom (Å) for chain C and 0.5 Angstrom (Å) for all other chains. The averaged coordinate structure was saved into a standard PDB file format. Table 1, Table 2 and Table 3 show the number of residues averaged per chain for each target. Target T201 has 73.55% residues averaged, target T202 has 86.60% residues averaged, and finally target T203 has 57.70% residues averaged.
**Algorithm 1** Average predicted structure and reference structure.**Require:** threshold ▹ threshold can be modified per chain  1: compute 
distance
 using Equation (Equation 1)  2: initialize: 
$x_{avg}=0,\;y_{avg}=0,and\;z_{avg}=0$
  3: **if** 
distance<threshold
 **then**  4:     
$x_{avg}⇐\left(x_{r}+x_{p}\right)/2$
  5:     
$y_{avg}⇐\left(y_{r}+y_{p}\right)/2$
  6:     
$z_{avg}⇐\left(z_{r}+z_{p}\right)/2$
  7: **else**  8:     
$x_{avg}⇐x_{r}$
  9:     
$y_{avg}⇐y_{r}$
 10:     
$z_{avg}⇐z_{r}$
 11: **end if**

After the backbone atoms were computed using Algorithm 1, we utilized SCWRL4 [40] to add the side-chain conformation into the protein structure. The deep learning-based method utilized to predict the backbone atoms had a high impact on determining the side-chains conformation as well, because high side chain accuracy is often achieved when the backbone prediction is accurate, as also demonstrated by AlphaFold [33].

### 2.2. Template-Based Prediction of Protein-Ligand Interaction

After the protein structure that can accommodate ligands was generated using Algorithm 1, we utilized PyRosetta [41] to identify ligands and add them into the predicted structure by using the reference structure as a template, as depicted in Algorithm 2. The reference structure contains the ligands’ atomic coordinates. Since PyRosetta is a residue based tool, when a pose is created, all the atoms in a structure including ligand atoms are indexed by residue indices. Following Algorithm 2, we let *res* be each residue in the reference structure that we checked for whether it was a ligand. PyRosetta’s *is_ligand* function works by comparing the ligand to a chemical component dictionary and returns a bool value (i.e., either True for ligand or False for non ligand) for each residue.
**Algorithm 2** Identify ligands and include them into average structure.**Require:** pyrosetta 1: initialize 
pyrosetta
 2: 
pose_ref=pose_from_pdb(referencestructure)
 3: **if** 
pose_ref.residue(res_id).is_ligand()==True
 **then** 4:     **with** 
open(“average_structure.pdb”,“a”)asfile:
 5:           
file.write(residue)
 6: **else** 7:     
donothing
 8: **end if**

### 2.3. Refinement of Protein-Ligand Complex Model

After the prediction of the protein–ligand complex structure using the approach outlined above, we further refined the predicted complex structure using Rosetta FastRelax. Relax does not perform extensive refinement and only searches the local low-energy backbone and side-chain conformations near the starting conformations by implementing rounds of packing and minimizing, with repulsive weight in the scoring function gradually increasing from a low value to a normal value. The scoring function we used was *ref2015_cst.wts*, which is a default score function, repeated five times. Finally, after the refinement of the protein complex, we used UCSF Chimera’s *Fit in Map* function to perform a rigid body optimization of the refined model. The 3D structure was rotated and aligned so that it fit to the density map. This refinement step was optional. During the blind experiment of the 2021 EMDataResource Ligand Challenge, we submitted both an unrefined model and a refined model for each target.

### 2.4. Target cryo-EM Density Maps of 2021 Ligand Challenge

We blindly tested the protein–ligand modeling pipeline DeepProLigand on three targets that were released as 2021 EMDataResource Ligand Challenge targets from February to April 2021. The next section elaborates the three targets and the experimental setting used for each target.

#### 2.4.1. Target 201: *Escherichia coli* Beta-Galactosidase

The 
β
-Galactosidase [42] target with atomic resolution of 1.9 Angstrom (Å) contains protein Beta-galactosidase, magnesium ion, sodium ion, water and 2-phenylethyl 1-thio-beta-D-galactopyranoside (PTQ) as a ligand. The EMDB ID of the target in EMDataResource is EMD-7770. We predicted the 3D structure of the complex using the workflow of DeepProLigand, as highlighted in Figure 2 and, during averaging of the structure, we initialized a 0.3 Angstrom (Å) distance threshold for chain C and a 0.5 Angstrom (Å) distance threshold for all other chains of the complex by re-initializing the threshold value of Equation (Equation 2). The reason for threshold of 0.3 Å in chain C was because most of the chains were turned into coils/turns with a threshold of 0.5 Å. The ligand PTQ was appended using Algorithm 2. Figure 3 shows the map–model overlay of cryo-EM density map EMD-7770 and our reconstructed protein structure model.

#### 2.4.2. Target 202: SARS-CoV-2 RNA-Dependent RNA Polymerase

The nsp12-nsp7-nsp8 complex bound to the template-primer RNA and triphosphate form of Remdesivir(RTP) [43] target with an atomic resolution of 2.5 Angstrom (Å) contains RNA-directed RNA polymerase, Non-Structural Protein 8, Non-Structural Protein 7, Primer, Templete, ZINC ION, PYROPHOSPHATE 2-, MAGNESIUM ION, water, and [(2 R,3 S,4 R,5 R)-5-(4-azanylpyrrolo[2,1-f][1,2,4]triazin-7-y)-5-cyano-3,4-bis(oxidanyl)oxolan-2-yl]methyl dihydrogen phosphate as a ligand. The EMDB ID of the target in EMDataResource is EMD-30210. We predicted the 3D structure of the complex using the workflow of DeepProLigand as highlighted in Figure 2 and, during averaging of the structure, we used a 1 Angstrom (Å) distance threshold for all chains of the complex. The ligand F86 (remdesivir, covalent inhibitor) was appended using Algorithm 2. Figure 4 shows the map–model overlay of cryo-EM density map EM-30210 and our reconstructed protein structure model.

#### 2.4.3. Target 203: SARS-CoV-2 Protein 3a in Lipid Nanodiscs

The SARS-CoV-2 3a ion channel in lipid nanodiscs [35] target with atomic resolution of 2.08 Angstrom (Å) contains ORF3a protein, Apolipoprotein A-I, water and 1,2-Dioleoyl-sn-glycero-3-phosphoethanolamine as a ligand. The EMDB ID of the target in EMDataResource is EMD-22898. We predicted the 3D structure of the complex using the workflow of DeepProLigand as highlighted in Figure 2 and, during averaging of the structure, we used a 1 Angstrom (Å) distance threshold for all chains of the complex. The ligand PEE was appended using Algorithm 2. Figure 5 shows the map–model overlay of cryo-EM density map EMD-22898 and our reconstructed protein structure model. The source code, data, and instructions to reproduce the results are available in the GitHub repository: https://github.com/jianlin-cheng/DeepProLigand, accessed on 8 January 2023.

## 3. Results

The analysis of the models in this section is based on the official results provided by the organizers of the 2021 Ligand Model Challenge. The fit to a map for a ligand was assessed by the Q-score [44] and the Z-scores. The Q-score measures how similar map values around an atom are to a Gaussian-like function which we would see if the atom were well resolved. The Q-score was calculated as a correlation between two vectors: u, which contained map values at points around the atom, and v, which contained values obtained from the reference Gaussian.

We used the Q-score to compare the map-to-model fit for all the models that were submitted to the challenge. Table 4 shows the Q-score of the ligand for all the models submitted for Target 201; our model is highlighted in bold for scrutiny. Figure 6 shows the ligand (PTQ)’s binding pose and orientation in our best predicted model, T0201EM004_1. Ligand PTQ bound to all four chains of Target 201, resulting in four binding sites for the ligand. We visualized three binding locations for the ligand with its binding pose and orientations in Figure 6. Table 5 shows the Q-score of the ligand for Target 202 and, similar to Target 201, our model is highlighted in bold for scrutiny. Figure 7 shows ligand (F86)’s binding pose and orientation in our best predicted model, T0201EM004_1. Ligand F86 bound to only one location in Target 202. We visualized the binding location for the ligand with its binding pose and orientations in Figure 7.

Table 6 shows the Q-score of the ligand for Target 203. Similar to Target 201 and 202, our model is highlighted in bold for scrutiny in Table 6. Figure 8 shows the ligand (PEE)’s binding pose and orientation in our best predicted model, T0201EM004_1. Ligand PEE bound to two locations in Target 203. We have visualized the binding locations for the ligand with its binding pose and orientation in Figure 8.

Figure 9 shows cumulative Z-scores on Q-scores of 17 groups participating in the 2021 Ligand Model Challenge. Our DeepProLigand predictor (EM004) performs best overall on all three targets. Specifically, our protein–ligand model was ranked first for Target 202, second for Target 203, and in the middle for Target 201 as shown in the Z-scores on Q-scores in Figure 9. The Q-scores of our best model for the three targets are shown in Table 7. Even though there are too few targets to draw a definite conclusion, the good results indicate that the deep learning structure prediction in conjunction with the template reference structure is able to build a good protein structure framework to accommodate ligands, and the template-based protein–ligand prediction can assemble the ligands with the protein structure well for some targets. Incorporating a deep learning approach in modeling enables us to predict the protein structure directly from the cryo-EM map within minutes, making the approach highly useful in terms of both prediction accuracy and time.

We also noticed the limitation of our approach in terms of the geometric quality of the atoms in the predicted protein structure, however. Particularly, there were some atom–atom clashes in the models, which may be caused by the violations of some geometric constraints of atom–atom distances in the protein structure predicted by the deep learning, as well as in the averaging of the coordinates of the predicted structure and the reference structure. The violations of geometric and stereochemical restraints were not fixed by the current refinement protocol in the prediction pipeline. The refinement protocol even introduced some new clashes into the model. AlphaFold demonstrated that the well-trained sophisticated deep learning architecture can accurately capture the geometric restraints of atoms and bonds in protein structures by predicting high-quality protein structures of atomic resolution that are highly similar to natural protein structures; this means more advanced deep learning architectures can be developed to predict high-quality protein structures compatible with the geometric and stereochemical restraints of proteins from cryo-EM density maps and reference structures.

## 4. Conclusions and Future Work

In this work, we demonstrate that the deep learning prediction of protein structures from cryo-EM maps can generate good protein structures for constructing protein–ligand complexes and the template-based protein–ligand interaction prediction can fit ligands well into the predicted protein structures according to the outstanding performance of our protein–ligand modeling pipeline. It is also worth noting that our method was fully automatic and did not involve any manual tweaking of the models to improve the scores. As discussed before, the current protein–ligand prediction pipeline cannot resolve some violations of some geometric and stereochemical restraints of atoms in protein structures. We plan to soon develop advanced end-to-end deep learning architectures, similar to some components in AlphaFold, to better predict better protein structures from cryo-EM maps and reference structures. Moreover, we plan to design 3D-equivariant deep learning architectures like the SE(3)-equivariant Transformer network [45,46,47,48] to tackle the problem of geometric constraints which are not addressed by current methods. Finally, an end-to-end direct deep learning prediction of the structure of protein–ligand complexes from cryo-EM density maps, reference structures and ligand information to fully automate all the steps of the entire pipeline in this work will be pursued. We believe the application of a deep learning approach to the prediction of 3D structures of protein–ligand complexes leveraging cryo-EM and other related data is a promising avenue with which to accelerate the advancement of the study of protein–ligand interaction [49,50]. With the proliferation of cryo-EM maps being deposited in the EMDataResource database, the use of deep learning-based methods can help to determine the structure of the protein–ligand complexes rapidly and ultimately help to expedite the drug discovery process.

## Figures and Tables

**Figure 2 biomolecules-13-00132-f002:**
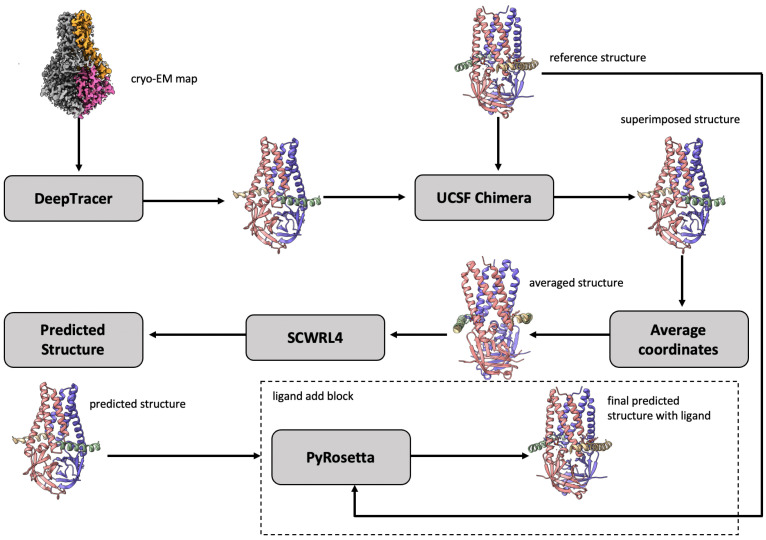
The workflow of DeepProLigand generating protein complex structure from cryo-EM map and reference structure. The cryo-EM map (EMD-22898) illustrated in the workflow is of a SARS-CoV-2 ORF3a ion channel in lipid nanodiscs [35].

**Figure 3 biomolecules-13-00132-f003:**
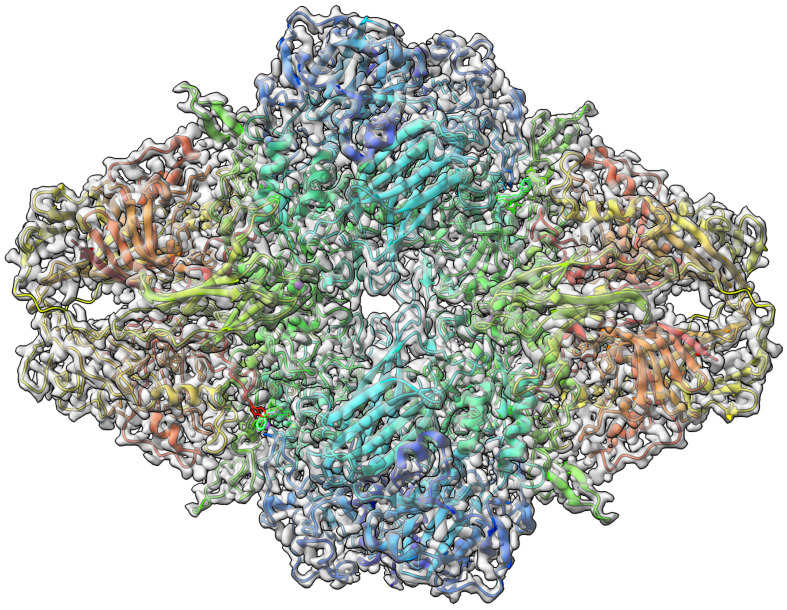
Target 201 (EMD-7770) map-model overlay at the recommended contour 0.52 (3.3 
σ
) with T0201EM0004_1 (ours).

**Figure 4 biomolecules-13-00132-f004:**
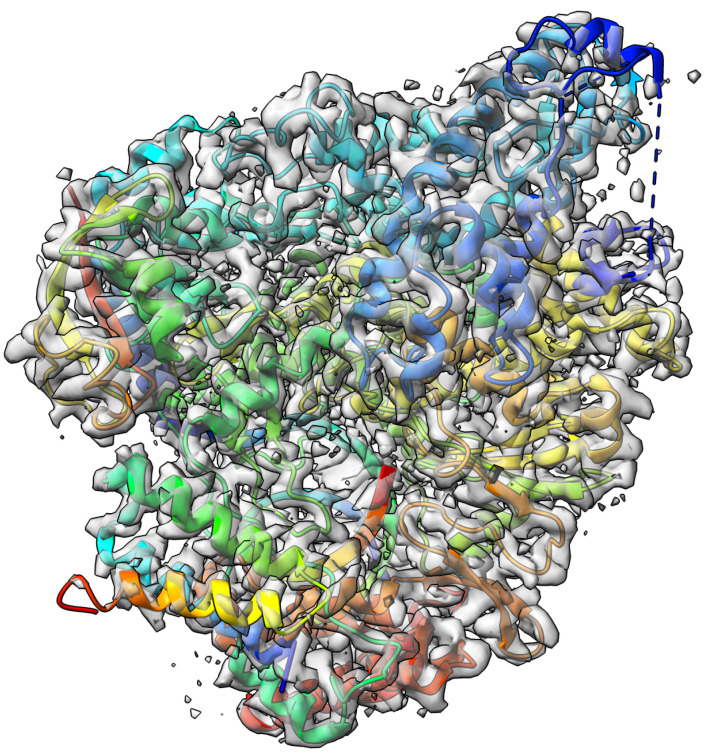
Target 202 (EMD-30210) map-model overlay at the recommended contour 0.058 (4.3 
σ
) with T0202EM004_1 (ours).

**Figure 5 biomolecules-13-00132-f005:**
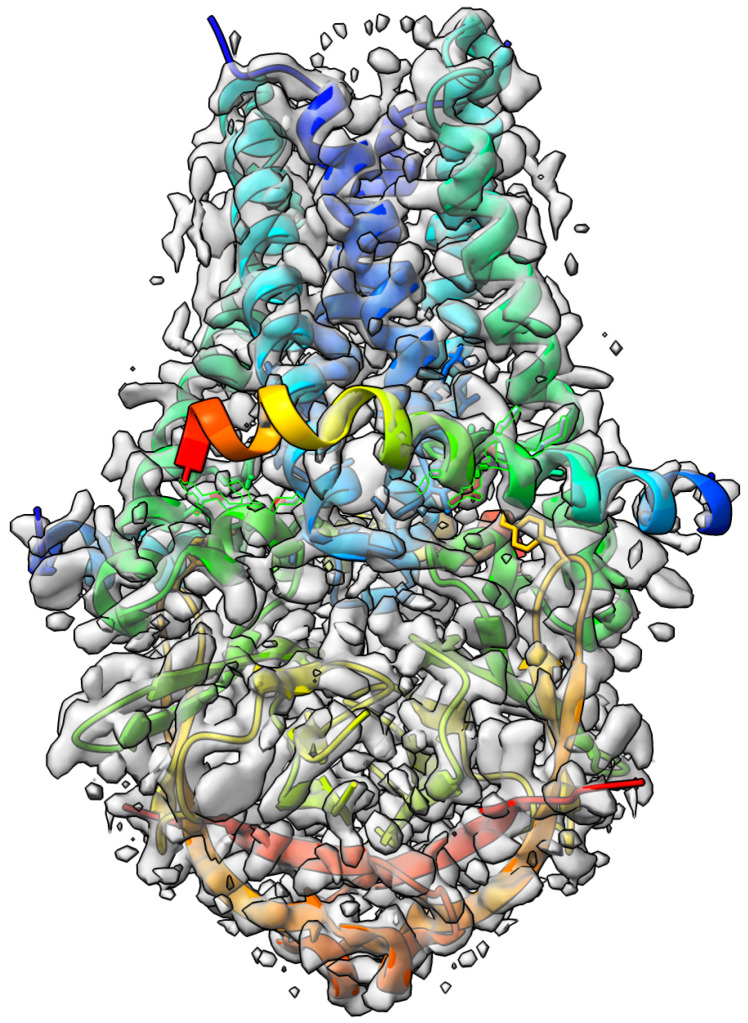
Target 203 (EMD-22898) map-model overlay at the recommended contour 0.7 (10.3 
σ
) with T0203EM004_1 (ours).

**Figure 6 biomolecules-13-00132-f006:**
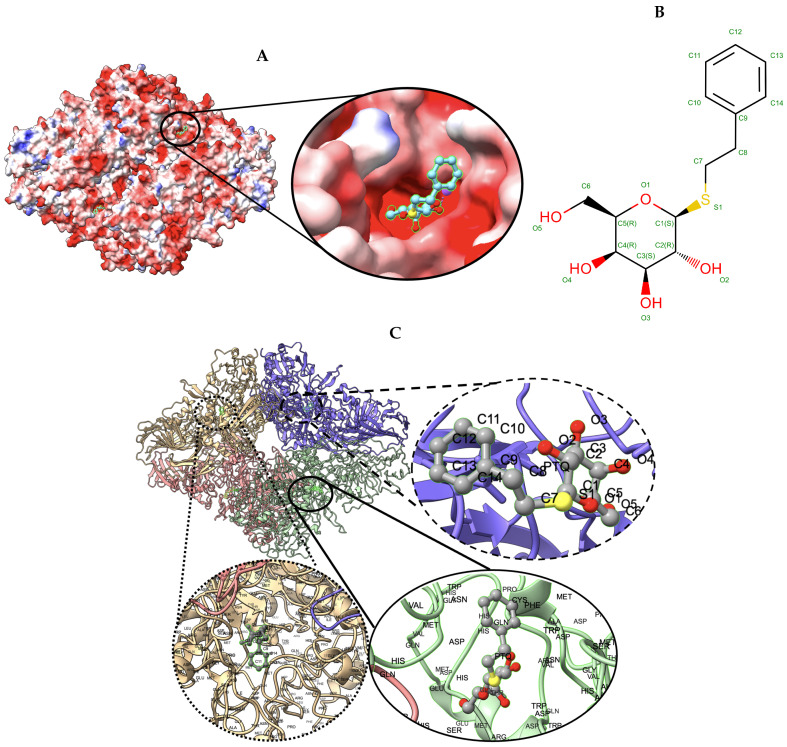
Target 201. (**A**) T0201EM004_1 (ours) docked by Target 201 (EMD-7770) and visualized with electrostatic potential surface generated in UCSF Chimera. (**B**) Ligand PTQ, image extracted from Protein Data Bank (PDB). (**C**) Protein–ligand interactions in T0201EM004_1 (ours) model. Chains are colored differently (chain A: blue, chain B: pink, chain C: green and chain D: golden). The ligand is labeled with its atom names as well as the ligand name (PTQ). For chain D: golden and chain C: green, we have labeled the chain residue names for understanding protein–ligand interaction better.

**Figure 7 biomolecules-13-00132-f007:**
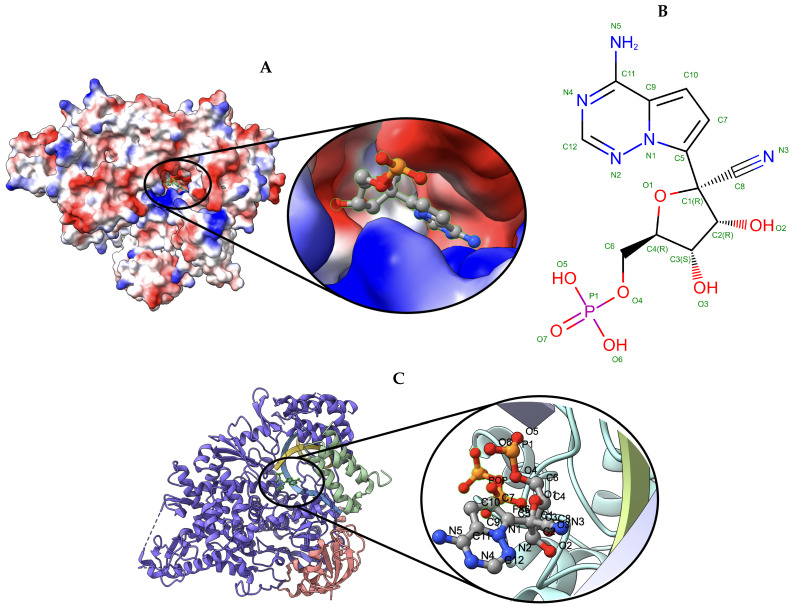
Target 202. (**A**) T0202EM0004_1 (ours) docked by Target 202 (EMD-30210) and visualized with electrostatic potential surface generated in UCSF Chimera. (**B**) Ligand F86, image extracted from Protein Data Bank (PDB). (**C**) Protein–ligand interactions in T0202EM004_1 (ours) model. Chains are colored differently (chain A: blue, chain B: orange, chain C: green, chain P: yellow, and chain T: teal). The ligand is labeled with its atom names as well as the ligand name (F86).

**Figure 8 biomolecules-13-00132-f008:**
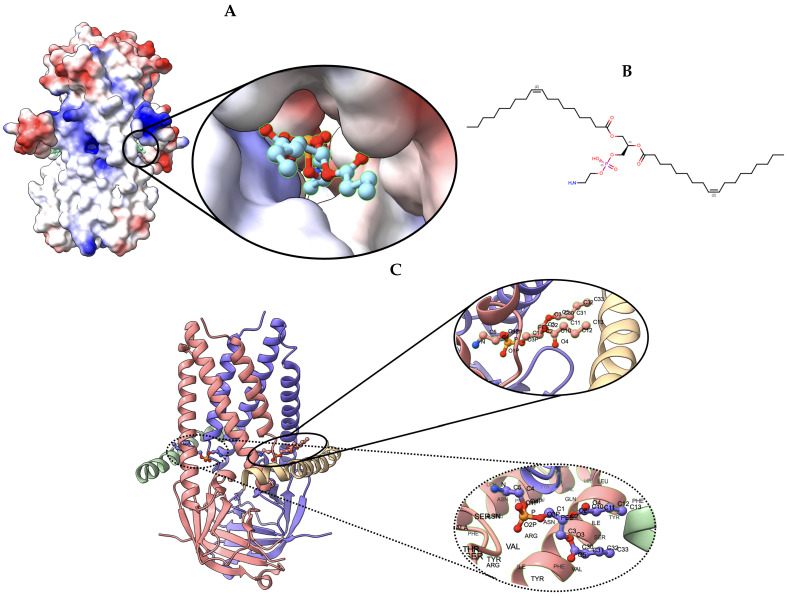
Target 203. (**A**) T0203EM0004_1 (ours) docked by Target 203 (EMD-22898) and visualized with electrostatic potential surface generated in UCSF Chimera. (**B**) Ligand PEE, image extracted from Protein Data Bank (PDB). (**C**) Protein–ligand interactions in T0203EM0004_1 (our) model. Chains are colored differently (chain A: blue, chain B: pink, chain C: green, and chain D: golden). The ligand are labeled with their atom names as well as the ligand’s name (PEE).

**Figure 9 biomolecules-13-00132-f009:**
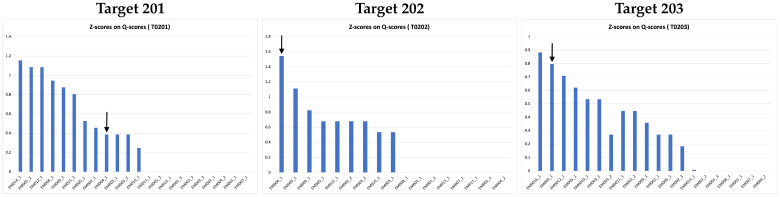
Z-scores on Q-scores for ligand of all the models submitted to 2021 Ligand Model Challenge. The pointed arrow represents our model.

**Table 1 biomolecules-13-00132-t001:** Number of residues averaged for target T201: EMD 7770.

Target T201: EMD 7770			
**Chain ID**	**Total Residues**	**Averaged Residues**	**% of Residues Averaged**
Chain A	1021	845	82.8
Chain B	1021	845	82.8
Chain C	1021	461	45.2
Chain D	1021	852	83.4
Average % across chains			73.55

**Table 2 biomolecules-13-00132-t002:** Number of residues averaged for target T202: EMD 30210.

Target T202: EMD 30210			
**Chain ID**	**Total Residues**	**Averaged Residues**	**% of Residues Averaged**
Chain A	834	762	91.4
Chain B	114	105	92.1
Chain C	63	48	76.2
Average % across chains			86.6

**Table 3 biomolecules-13-00132-t003:** Number of residues averaged for target T203: EMD 22898.

Target T203: EMD 22898			
**Chain ID**	**Total Residues**	**Averaged Residues**	**% of Residues Averaged**
Chain A	193	184	95.3
Chain B	193	0	0
Chain C	31	20	64.5
Chain D	31	22	71.0
Average % across chains			57.7

**Table 4 biomolecules-13-00132-t004:** Evaluation of Target 201: *Escherichia coli* beta-galactosidase on Q-score for all the models submitted in the 2021 Ligand Model Challenge.

Team Name	PTQ (Ligand)
T0201EM014_1	0.82
T0201EM005_2	0.81
T0201EM012_1	0.81
T0201EM006_1	0.79
T0201EM009_1	0.78
T0201EM015_1	0.77
T0201EM005_1	0.73
T0201EM002_2	0.72
**T0201EM004_1**	**0.71**
T0201EM003_1	0.71
T0201EM003_2	0.71
T0201EM010_1	0.69
T0201EM011_1	0.64
T0201EM001_2	0.64
T0201EM013_1	0.63
T0201EM001_3	0.62
T0201EM002_3	0.60
T0201EM003_3	0.58
T0201EM002_1	0.55
T0201EM001_1	0.33
T0201EM007_1	0.31
T0201EM008_1	-

Note: Table is sorted in descending order using ligand: PTQ score. “-” means, we were unable to calculate the score of the model. Our best model is highlighted in bold.

**Table 5 biomolecules-13-00132-t005:** Evaluation of Target 202: SARS-CoV-2 RNA-dependent RNA polymerase on Q-score for all the models submitted to the 2021 Ligand Model Challenge.

Team Name	F86 (Ligand)
**T0202EM004_1**	**0.74**
T0202EM009_1	0.71
T0202EM006_1	0.69
T0202EM005_1	0.68
T0202EM012_1	0.68
T0202EM002_2	0.68
T0202EM003_2	0.68
T0202EM010_1	0.67
T0202EM003_1	0.67
T0202EM008_1	0.63
T0202EM001_1	0.60
T0202EM001_2	0.59
T0202EM013_1	0.59
T0202EM007_1	0.57
T0202EM011_1	0.56
T0202EM002_1	0.52

Note: Table is sorted in descending order using ligand: F86 score. Our best model is highlighted in bold.

**Table 6 biomolecules-13-00132-t006:** Evaluation of Target 203: SARS-CoV-2 ORF3a putative ion channel in nanodisc on Q-score for all the models submitted in the 2021 Ligand Model Challenge.

Team Name	PEE (Ligand)
T0203EM0016_1	0.77
**T0203EM004_1**	**0.76**
T0203EM0012_1	0.75
T0203EM005_1	0.74
T0203EM0010_1	0.73
T0203EM003_1	0.73
T0203EM003_2	0.70
T0203EM0011_1	0.72
T0203EM002_2	0.72
T0203EM009_1	0.71
T0203EM002_1	0.70
T0203EM006_1	0.70
T0203EM002_3	0.69
T0203EM0014_1	0.67
T0203EM001_2	0.66
T0203EM001_3	0.63
T0203EM008_1	0.63
T0203EM001_1	0.60
T0203EM007_1	0.51

Note: Table is sorted in descending order using ligand: PEE score. Our best model is highlighted in bold.

**Table 7 biomolecules-13-00132-t007:** Q-score of our best model (EM004_1) for three targets.

Target Name	Ligand
Target 201	0.71 (PTQ)
Target 202	0.74 (F86)
Target 203	0.76 (PEE)

Note: Among two models submitted for each target in the challenge, our best model’s id is EM004_1 across all the targets. EM004_1 model is the non refined model.

## Data Availability

The cryo-EM density maps are available in the EMDB website [2] with ID: EMD-7770, EMD-30210, and EMD-22898. The submitted models metadata and further detailed information on 2021 Ligand Model Challenge can be accessed from the official challenge web-page [3]. The protein complex generated from this method, DeepProLigand, and the software program are released, and can be accessed through the GitHub repository: https://github.com/jianlin-cheng/DeepProLigand, accessed on 20 November 2022.

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
