# Peer review of "Improving Protein–Ligand Interaction Modeling with cryo-EM Data, Templates, and Deep Learning in 2021 Ligand Model Challenge"

_biomolecules, 2023, doi:10.3390/biom13010132_

Round 1

Reviewer 1 Report

Reviewer’s Comments:

The manuscript “A Deep Learning Bioinformatics Approach for Modeling Protein-Ligand Interaction with cryo-EM Data in 2021 Ligand Model Challenge” is a very interesting work. Herein, authors developed a deep learning bioinformatics pipeline (DeepProLigand) to predict protein-ligand interactions from cryo-EM density maps of proteins and ligands. DeepProLigand first uses a deep learning method to predict the structure of proteins from cryo-EM maps, which is averaged with a reference (template) structure of the proteins to produce a combined structure to add ligands. The ligands are then identified and added into the structure to generate a protein-ligand complex structure, which is further refined. The method based on the deep learning prediction and template-based modeling was blindly tested in the 2021 EMDataResource Ligand Challenge and was ranked first in fitting ligands to cryo-EM density maps. The results demonstrate the deep learning bioinformatics approach is a promising direction to model protein-ligand interaction on cryo-EM data using prior structural information.  The results are consistent with the data and figures presented in the manuscript. While I believe this topic is of great interest to our readers, I think it needs major revision before it is ready for publication. So, I recommend this manuscript for publication with major revisions.

1. In this manuscript, the authors did not explain the importance of the Protein-Ligand Interaction in the introduction part. The authors should explain the importance of Protein-Ligand Interaction.

2) Title: The title of the manuscript is not impressive. It should be modified or rewritten it.

3) Correct the following statement “The source code, data, and instruction to reproduce the results are open sourced and available on GitHub repository : https://github.com/jianlin-cheng/DeepProLigand”.

4) Keywords: The EMDataResource ligand model challenge is too long. So, modify the keywords.

5) Introduction part is not impressive. The references cited are very old. So, Improve it with some latest literature such as 10.1016/j.inoche.2022.109702, 10.3389/fmats.2022.875163

6) The authors should explain the following statement with recent references, “The results demonstrate the deep learning bioinformatics approach is a promising direction to model protein-ligand interaction on cryo-EM data using prior structural information. The source code, data, and instruction to reproduce the results are open sourced and available on GitHub repository : https://github.com/jianlin-cheng/DeepProLigand”.

7) Add space between magnitude and unit. For example, in synthesis “21.96g” should be 21.96 g. Make the corrections throughout the manuscript regarding values and units.

8) The author should provide reason about this statement “The violations of geometric and stereochemical restraints are not fixed by the current refinement protocol in the prediction pipeline”.

9. Comparison of the present results with other similar findings in the literature should be discussed in more detail. This is necessary in order to place this work together with other work in the field and to give more credibility to the present results.

10) Conclusion part is very long. Make it brief and improve by adding the results of your studies.

11) There are many grammatic mistakes. Improve the English grammar of the manuscript.

Reviewer 2 Report

The paper by Giri and Cheng developed a pipeline for predicting ligand protein interactions from the cryoEM maps. This pipeline achieved a good accuracy by taking advantage of existing structure prediction and refinement software including DeepTracer, SCWRL4 and PyRosetta.

The major problem of this paper is that the author did not describe how the reference structure is provided, whether it is from manually built atomic model, the alpha-fold predicted model or the ground truth structure from PDB?  It seems that this reference structure already have accurate coordinates of both protein and ligand, what are the scores for ligand in this reference structure. If the reference structure is from PDB or provided by the 2021 Ligand Model Challenge, the atomic model of proteins should be carefully built by experts so that DeepTracer should not improve atomic modeling accuracy. In this respect, the only valuable step in this pipeline is to refine ligand with Rossetta, which is not novel here. If the reference structure is not from PDB, please describe how it is generated.

Other points of concerns are listed below:

1.       Line 50 “deep learning-based architecture as a core component of our modeling pipeline”. My understanding is that the author refers DeepTracer as deep learning-based architecture. If that is the case, it would be better to acknowledge it explicitly.

2.       The message in Line 99-100, 105-106 about distance threshold is repeated in Line 111-115.

3.       Line 130, “ligand residue” is technically incorrect, it should be “ligand” or “ligand binding residue”.

4.       The author claims their method ranked the first in fitting ligands to cryo-EM maps, however it is only the first in one metric of one target.  Please clarify it.

5.       The authors talked about the model geometry quality. Could you provide Rama, rotamers, clashes scores of the models?

Reviewer 3 Report

This manuscript describes a new approach that integrates deep learning to model protein-ligand complex structures. While the idea is exciting given the success of AlphaFold, I found there are a few issues to be addressed.

  1. First, the authors described an approach with a deep learning element, mostly from the DeepTracer tool they used. It seems inappropriate to have just “deep learning”, but not the other important element, i.e. template-based modeling, in the title. Additionally, the authors did not seem to provide strong evidence that it is their very deep learning tool that makes a significant difference (in comparison to other deep learning tools and/or traditional approaches).

  2. The term “interaction” is somewhat misleading. In the context of the protein-ligand complex, “interaction” could simply indicate they do interact or if they interact as opposed to how they interact geometrically. I believe the authors should make it more explicit for this manuscript, as they are describing a method to predict the bound conformation (3D atomic structure) of protein-ligand complexes.

  3. At Line 47, can the authors elaborate why “existing methods cannot leverage vast structural data effectively” in the context of protein-ligand bound structures?

  4. The success of AlphaFold is not simply due to the deep learning algorithm, but also the much larger amount of (non 3D structure) data used. The authors should not ignore the latter in their introduction.

  5. The term “protein complex” is also misleading as there are many protein-protein complexes. It is better to be more specific as the authors are studying the “protein-ligand complex” in this manuscript.

  6. It appears that the authors chose DeepTracer in the first step because it utilizes both the cryoEM map and the amino acid sequence. Would AlphaFold predictions that do not utilize cryoEM maps but amino acid sequences give better or worse results? Would the “traditional approach of fitting a reference structure into a cryo-EM density map” give better or worse results?

  7. Where does the “reference structure” come from in the workflow shown in Figure 2? Does this reference structure only have backbone atom coordinates? Does it have ligand coordinates? This last question is of particular importance as the main point of this manuscript is good prediction of ligand bound (conformational) structure.

  8. According to the authors’ Algorithm 1, the “averaged structure” could simply be from the reference structure, already known, depending on the distance threshold. Can the authors make it explicitly clear, in their three examples, that actual averaging was performed as opposed to directly taking backbone coordinates from the reference structure? Otherwise, there seems to be no deep-learning element in their workflow. In addition, why not just average without considering a distance threshold that seems arbitrary?

  9. The addition of ligands by PyRosetta might be the most important step of this study and I believe it deserves more attention. Does the “averaged structure” already contain information, at least residue label, for the ligand? If so, where does the ligand label come from? This also goes back to my earlier question of whether the referenced structure already has ligands.

  10. Finally, do any of the other teams in the 2021 EMDataResource Ligand Challenge use a docking-based method? Despite the authors’ suggestions, there seems to be no evidence in this manuscript showing their approach is indeed better than a workflow that uses traditional methods that fit a reference structure into a cryo-EM density map followed by molecular docking (with some restraints from the cryo-EM map).

Round 2

Reviewer 2 Report

I am satisfied with other points but please comment on my previous question: "If the reference structure is provided by the 2021 Ligand Model Challenge, the atomic model of proteins should be carefully built by experts so that DeepTracer should not improve atomic modeling accuracy. In this respect, the only valuable step in this pipeline is to refine ligand with Rossetta, which is not novel here."

In other words, you have a ground truth reference model, and you are constructing another model based on this already reliable ground truth with deep tracer and rossetta. Isn't it more reasonable to start with a bad reference?

Reviewer 3 Report

The authors have addressed many of the issues I raised previously and improved their manuscript in this second version, but there are several outstanding issues.

  1. Line 28, I thought the authors meant “new drugs” instead of “new drug targets”

  2. Line 52, “thousands of non 3D structure data” is far from accurate, and I suggest the authors have a careful read of the AlphaFold paper to improve their understanding.

  3. There are existing methods that leverage structural data, at least from crystallography, although they may not be regarded as “vast”. For example, some deep-learning (DL) based methods used PDBbind data, which consist of over 20,000 entries of protein-ligand structures. Some DL-based methods even went beyond prediction of ligand-bound conformation to also include ligand binding affinity. The authors should, at least, briefly discuss such prior work in the Introduction section.

  4. The authors responded to my previous Point 7 that “The reference structure contains ligand coordinates”, but I cannot seem to find a similar statement in the revised manuscript. Given the importance of ligand information, I strongly suggest the authors clarify the exact nature of available ligand information in the manuscript. In particular, are there really atomic “coordinates” being provided?

  5. The authors provided tables showing information of averaged residues in response to my previous Point 7. This is helpful and should be included in their manuscript, or at least in a supplementary file. The same goes for the two figures the authors provided in response to my previous Point 5.

  6. The authors should be up-front about potential caveats of this study. For example, they did not carry out a direct comparison of their approach vs a workflow that uses traditional methods that fit a reference structure into a cryo-EM density map followed by molecular docking (with some restraints from the cryo-EM map), and can only speculate their approach might be better given their success in the 2021 EMDataResource Ligand Challenge.

Round 3

Reviewer 2 Report

I am satisfied with the author response and do not have further questions.

Author Response

Thank you. 

Reviewer 3 Report

Although the authors improved their manuscript to some extent again, it is disappointing that they have not addressed some of the key questions I asked in the previous round of comments and appear to be ignoring these key issues. I’d like to raise, or rather, reiterate them one last time and hope the authors could carefully consider these points.

There are existing methods that leverage structural data, at least from crystallography, although they may not be regarded as “vast”. For example, some deep-learning (DL) based methods used PDBbind data, which consist of over 20,000 entries of protein-ligand structures. Some DL-based methods even went beyond prediction of ligand-bound conformation to also include ligand binding affinity. The authors should, at least, briefly discuss such prior work in the Introduction section.

I’d also like to point out that these DL-based methods for protein-ligand interactions have been developed even before the publication of AlphaFold.

Given the importance of ligand information, I strongly suggest the authors clarify the exact nature of available ligand information in the manuscript. In particular, are there really atomic “coordinates” being provided?

The authors seem to have confirmed that atomic coordinates of the ligand are indeed provided in this 2nd revision. If so, why not just take the atomic coordinates and directly convert them to the corresponding 3D structure? Why bother starting from these atomic coordinates and using alternative approaches such as PyRosetta + Rosetta FastRelax or molecular docking?
